# Phase transformation strengthening of high-temperature superalloys

T.M. Smith[1], B.D. Esser[1], N. Antolin[2], A. Carlsson[3], R.E.A. Williams[1], A. Wessman[4], T. Hanlon[5], H.L. Fraser[1], W. Windl[2], D.W. McComb[1] & M.J. Mills[1]

Decades of research has been focused on improving the high-temperature properties of nickel-based superalloys, an essential class of materials used in the hot section of jet turbine engines, allowing increased engine efficiency and reduced $CO_2$ emissions. Here we introduce a new 'phase-transformation strengthening' mechanism that resists high-temperature creep deformation in nickel-based superalloys, where specific alloying elements inhibit the deleterious deformation mode of nanotwinning at temperatures above 700 °C. Ultra-high-resolution structure and composition analysis via scanning transmission electron microscopy, combined with density functional theory calculations, reveals that a superalloy with higher concentrations of the elements titanium, tantalum and niobium encourage a shear-induced solid-state transformation from the γ′ to η phase along stacking faults in γ′ precipitates, which would normally be the precursors of deformation twins. This nanoscale η phase creates a low-energy structure that inhibits thickening of stacking faults into twins, leading to significant improvement in creep properties.

[1] Center for Electron Microscopy and Analysis, The Ohio State University, Columbus, Ohio 43212, USA. [2] Department of Materials Science and Engineering, The Ohio State University, Columbus, Ohio 43210, USA. [3] FEI Company, Achtserweg Noord 5, Eindhoven 5651, The Netherlands. [4] G.E. Aviation, Cincinnati, Ohio 45215, USA. [5] G.E. Global Research Center, Niskayuna, New York 12309, USA. Correspondence and requests for materials should be addressed to T.M.S. (email: smith.5881@osu.edu).

The relentless drive for energy efficiency in power generation and propulsion places development of high-performance materials at the forefront of materials science. Turbine engine efficiency and reduction in carbon emissions are directly related to engine operating temperature. With increasing temperatures, materials start to plastically deform under load, a process known as creep, which eventually sets the most severe limits on materials performance[1]. Therefore, increased performance in aircraft engines and land-based power generators require the development of a new generation of high-temperature structural materials that are resistant to creep. Among these materials, Ni-based superalloys offer a unique combination of creep, fatigue and corrosion resistance[1]. Superalloys have a face centred cubic (fcc), solid solution matrix ($\gamma$ phase) with coherent precipitates ($\gamma'$ phase) of the $Cu_3Au$ structure which constitute around 50 volume per cent of the microstructure. The $\gamma'$ phase provides superb resistance against shearing via lattice dislocation movement, and thus remarkable strength at temperatures as high as 700 °C—a crucial capability for turbine disk components.

Currently three different strengthening mechanisms are understood and used to improve the high-temperature performance of alloys: solid solution strengthening, precipitation hardening and grain boundary strengthening. Previous studies have explored how to maximize the potential from all three of these 'classical' strengthening mechanisms. Since the characterization of the $\gamma'$ phase in Ni-base superalloys by Bradley and Taylor in 1937 (refs 2,3), the development of high-temperature alloys has largely proceeded in incremental fashion, with new progress focusing directly on the shortcomings of the previous generation of alloys. Understanding the effect of specific elements in the compositionally complex superalloys remains a qualitative and highly empirical endeavour. While significant advances have been made in the prediction of microstructures and phase stability based on thermodynamic and kinetic databases[4–6], the ability to predict consequent mechanical properties for a given alloy and microstructure persists as a major challenge for the materials genome initiative[7]. A significant obstacle to computationally-directed high-temperature alloy development is the lack of quantitative, comprehensive understanding of deformation mechanisms controlling high-temperature behaviour for various alloy compositions, temperatures and applied stresses.

A primary goal of the present research is to provide quantitative insight into the effect of various alloying elements on the operative deformation mechanisms under conditions that are relevant to advanced engine designs, and in alloys that are closely related to those presently used for advanced turbine disk applications. This has been achieved by application of integrated computational materials science and engineering involving the coupling of aberration-corrected atomic-resolution imaging with state-of-the-art energy-dispersive X-ray (EDX) spectroscopy, and density functional theory (DFT) calculations. This coupled study has resulted in the discovery of a high-temperature strengthening mechanism which we refer to as 'phase transformation strengthening.' The identification of this mechanism and the accompanying mechanistic insights could enable advances in high-temperature alloy design.

## Results

**Mechanical testing and deformation analysis.** To demonstrate the effect of the new strengthening mechanism, we examine two similar Ni-base superalloys, ME3 and ME501, for which the main difference important for our purposes is the amount of $\eta$ phase formers (Nb, Ta, W, Hf, Ti), which is 9.1 wt% for ME3 and 13% for ME501 (see 'Methods' section, Supplementary Fig. 1, Supplementary Table 1 and Supplementary Note 1 for complete information on the two alloys). Figure 1 shows the compression creep response for ME3 and ME501 at 760 °C for the [001] orientation, that is, the time-dependent plastic strain at constant load. Minimizing these plastic strains is critical to the high dimensional stability required of turbine engine disk materials. The creep curves in Fig. 1a reveal the remarkably improved creep resistance of ME501 compared with ME3 at 760 °C and 552 MPa (the green and blue curves, respectively). For accurate evaluation of the deformation mechanisms between the two alloys, the ME3 compression creep stress was repeated at 414 MPa to obtain more comparable strain rates (red curve). Post-creep STEM analysis revealed for both alloys the presence of dislocations with Burgers vectors of the type $\frac{1}{2}<110>$ dislocation in the $\gamma$ matrix, and faulting in the $\gamma'$ precipitates, as can be seen in the [001] zone axis bright-field (BF) images shown in Fig. 1b,c for ME3 and ME501, respectively. High resolution, high-angle annular dark field

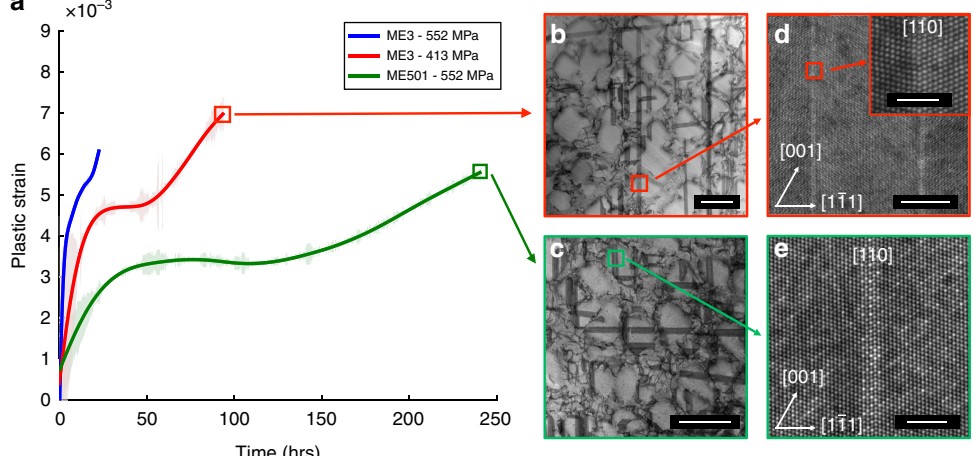

**Figure 1 | Compression creep data and deformation mechanisms. (a)** [001] Compression creep curves of both ME3 and ME501 at 760 °C and 552 MPa (blue and green, respectively) and ME3 at 414 MPa (red) to achieve comparable strain rates ($5 \times 10^{-9}\,s^{-1}$) between the two alloys. [001] Zone axis BF-STEM image revealing isolated SESFs and nanotwins in post-crept [001] **(b)** ME3 and **(c)** ME501 crystals. HAADF–STEM images showing **(d)** a nanotwin in ME3 and **(e)** a SESF in ME501. The inset in **(d)** shows a higher magnification of the twin interface in ME3. Scale bars in **b,c**, 500 nm. Scale bar in **d**, 5 nm (inset, 2 nm). Scale bar in **e**, 2 nm.

(HAADF) STEM shows that the faults extending through both the $\gamma$ and $\gamma'$ phases observed in ME3 were nanotwins, as exemplified in Fig. 1d. However, these nanotwins were absent in deformed ME501, while many instances of isolated stacking faults could be seen, which were constrained to the $\gamma'$ phases. High resolution HAADF–STEM has also revealed that these isolated faults are extrinsic faults (that is, superlattice extrinsic stacking faults (SESFs)) as shown in Fig. 1e consisting of stacking faults on adjacent close-packed {111} planes of the $\gamma'$ phase.

Quantifying the propensity for these deformation modes with electron channelling contrast imaging (ECCI) and STEM analysis, we found that nanotwinning was a prominent deformation mode in ME3, contributing up to 50% of the accounted plastic strain, while it was not operative in ME501 (See Supplementary Fig. 2, Supplementary Tables 2 and 3 and Supplementary Note 2).

**Atomic-scale characterization of faults.** To understand these differences in deformation modes, we start by analyzing isolated SESFs using HAADF–STEM as shown in the <110> zone axis images for both ME3 and ME501 in Fig. 2a,b, respectively. In both cases, a distinct local composition at the faults is observed. As described in a preliminary study[8], the ordering observed along the fault in ME501 can be attributed to a shear-induced phase transformation from $\gamma'$ to $\eta$ phase along the fault. Bulk $\eta$ phase possesses a hexagonal $D0_{24}$ crystal structure ($P6_3/mmc$, $a = 0.5096$, $c = 0.8304$, $\alpha = \beta = 90°$, $\gamma = 120°$), the same found locally along a SESF[8,9]. In ME3, brighter contrast, attributed to a larger average atomic number at the SESFs is also observed; however, the lack of atomic ordering along the fault implies that a different segregation event has occurred[10–12].

In addition, the concentration profiles across the SESFs from vertically integrated EDX line scans also show distinct differences.

For ME501, enrichment of Nb, Ta and Ti to the fault is found (Fig. 2d)—elements that are known to favour the formation of $\eta$ phase, which is ordinarily considered to be a deleterious phase compromising strength and ductility when present as a micrometre length-scale phase[13]. The small amount of segregation needed to nucleate the $\eta$ phase along the SESF implies that a threshold may exist where $\eta$ phase is promoted over $\gamma'$. In fact, one study found that when the C content was lowered from 0.15 to 0.08 wt% in IN792, thereby increasing the content of carbide formers (Ta, Nb and Ti) in the bulk alloy, that $\eta$ phase would begin to form[13]. It may be that the composition of ME501 is approaching this threshold. In ME3, Cr and Mo is observed to segregate along the fault (Fig. 2c) which are known to be elements that favour the formation of the $\gamma$ phase[14].

In previous studies of single crystal Ni-based superalloys, it has been found that orientations which promote nanotwinning result in poor creep performance[15–18]. In fact, nanotwinning has been speculated as the source of the tension/compression anisotropy observed in single crystal creep tests for Ni-based superalloys, with the directions that encourage nanotwinning exhibiting inferior creep strength[19]. This is consistent with the poor creep strength exhibited in this study by ME3 compared with ME501, as nanotwinning is observed only in ME3. To more deeply understand the relationship between SESFs and nanotwinning, as well as the role of atomic arrangement, site-specific analysis of the distribution of the different elements is essential. The most commonly used method in alloy research for this task is EDX. However, due to the stochastic nature of signal generation, even in thin foils, the spatial resolution of EDX has been previously limited to the nanometre scale for superalloys due to interaction volume and large number of alloying elements present[20]. However, the recent development of advanced, high sensitivity

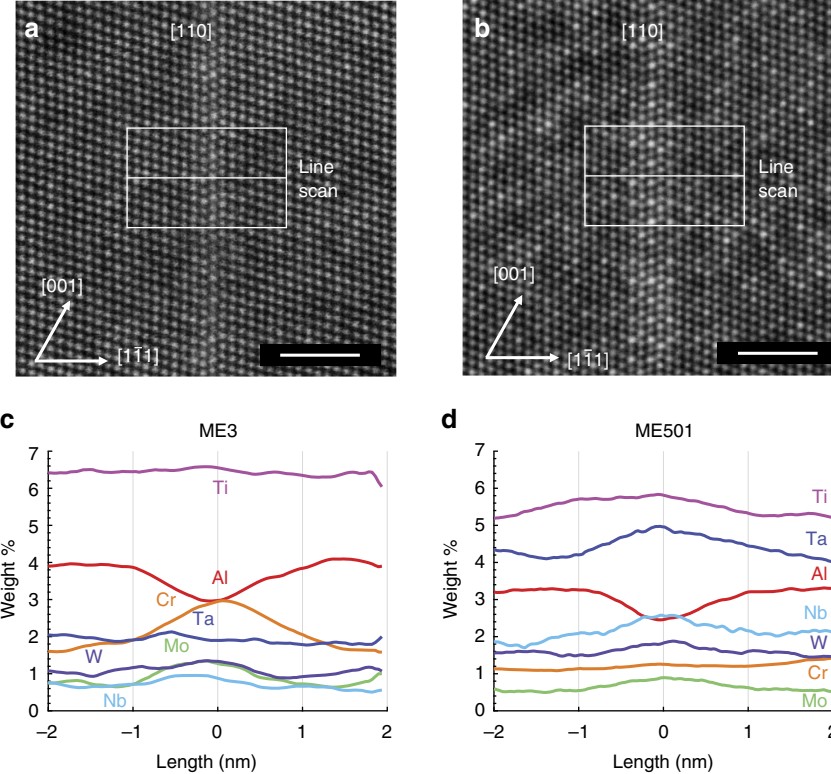

**Figure 2 | Segregation along stacking faults in ME3 and ME501.** HAADF–STEM image obtained on a [110] zone axis revealing (**a**) segregation along a SESF in ME3 and (**b**) segregation and 'grid-like' ordering of $\eta$ phase along a SESF in ME501. Integrated EDX line scans showing elemental segregation along a SESF for (**c**) ME3 and (**d**) ME501, as indicated in (**a,b**), respectively. Not shown is the enrichment of Co and the depletion in Ni content along the faults in both cases. Scale bars, 2 nm.

X-ray detector systems[21] has opened the door to use X-ray emission to characterize materials at previously unattainable spatial resolution. In fact, we demonstrate here for the first time in a structural metal alloy, that atomic resolution EDX maps can be obtained, as shown in Fig. 3, and provide quantified, site-specific segregation of solute atoms which definitively confirm that the η phase has formed locally at the stacking faults in ME501. Technical details about instrumentation and correction factors are provided in the Supplementary Figs 3–5, Supplementary Tables 4 and 5, and Supplementary Note 3.

At the stacking fault, it can be seen that Ta and Nb segregate preferentially to the indicated (circled) positions, which are the Wyckoff 2a positions of the bulk η phase (for the Nb map see Supplementary Fig. 6). Furthermore, Al and Ti are observed to segregate to the Wyckoff 2d positions of the bulk η phase, while Co can be seen to segregate to the Ni sublattice. Not only do these site-specific results account for the Z-contrast intensity observed in the HAADF–STEM images, they also confirm the assertion that the η phase has nucleated along the SESF, and matches the ordering first described by Pickering et al.[9] in a different alloy (718 plus), also containing elevated Ti, Nb and Ta content.

**Density functional theory calculations**. Taken together, Figs 2 and 3 indicate, for the first time that elemental segregation to stacking faults occurs differently in these two important, commercial alloys. Elemental segregation and nucleation of the η phase occurs along the faults in ME501. However, the segregation in ME3 indicates a distinctly different trend for forming a γ-like phase at the stacking faults, with Cr and Mo replacing Al. Several reports, this analysis included, have found nanotwinning to occur in orientations that favour SESF formation[17,19,22]. This correlation can be rationalized mechanistically by noting that additional shearing events by Shockley partial pairs will lead to thickening of stacking faults into nanotwins as shown below in Fig. 4.

To understand how the different segregation/phase transformation phenomena observed between the two alloys affect formation of the detrimental nanotwinning deformation mode, DFT calculations were performed on simulation cells such as those in Fig. 4a,c, created using knowledge gleaned from the site-specific EDX maps in Fig. 3 and HAADF–STEM images in Fig. 2, using the Vienna Ab-Initio Simulation Package (VASP)[23]

Figure 5a displays the energetics of the twin formation process in the Ni-based superalloys studied.

Once the SESF configuration has formed, it is necessary for two additional partial dislocations to shear adjacent planes to create a four-layer nanotwin configuration, as shown in Fig. 4; however, in the absence of diffusional rearrangement of the elemental segregation, this process creates a plane of atoms with high-energy, nearest-neighbor violations. We consider this configuration as an energy barrier for thickening into twins in the ordered γ′ superlattice: as shown in Fig. 5a, for pure Ni₃Al this barrier is $364 \, \text{mJ m}^{-2}$, for alloy ME501 where the solutes are distributed as a random solid solution (indicated as 'RSS') the barrier is $481 \, \text{mJ m}^{-2}$, and for the segregated SESF with η phase the barrier is $743 \, \text{mJ m}^{-2}$. It may be concluded that the ordered η phase observed at the SESFs in ME501 creates a significant barrier for nanotwin formation in this alloy system[8].

**Discussion**

Following the hypothetical shearing of planes adjacent to the SESF, atomic reordering is necessary to create low-energy nanotwins[24–26]. In order for a SESF and twin to form by movement of Shockley partial dislocations, a reordering process must occur to eliminate high-energy, wrong nearest-neighbor bonds along the fault and twins. Kolbe[26] and Kovarik et al.[24] both described the mechanism in which a fault created by the shear of like-signed Shockley partials on adjacent {111} planes in the γ′ phase can reorder by a local diffusional process and eliminate wrong nearest-neighbor violations. In Fig. 5b, where the energetics of this process are also considered, a large release of energy associated with these atomic rearrangements is found: for pure Ni₃Al, this release is on the same order as the additional energy for shearing of partials, that is, $375 \, \text{mJ m}^{-2}$. For the ME3 alloy case, this process releases almost no energy at all: only $15 \, \text{mJ m}^{-2}$, much less than the energy associated with twin formation. This reduction in energy in ME3 is a result of the loss of nearest-neighbor violations in the γ′ precipitate—a significant contributor to the high strength of superalloys—from the segregation of γ formers along the fault.

The implications from the DFT calculations in Fig. 5 are summarized in Fig. 6. In the case of ME3, γ formers (Co, Cr, Mo) segregate to the fault, transforming the fault to a γ-like region, as shown in Fig. 6b. New Shockley partials interact at the

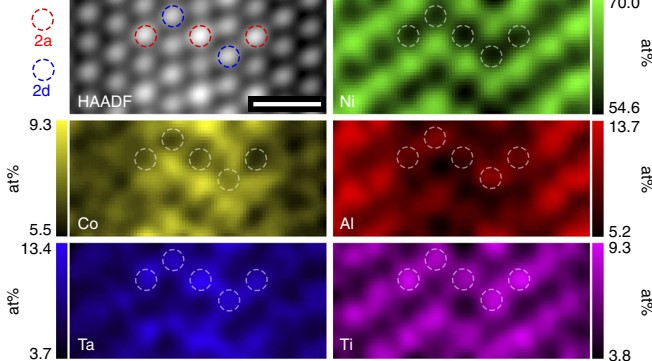

**Figure 3 | High spatial resolution elemental maps across a SESF in ME501.** Quantified atomic resolution EDX elemental maps at atomic scale of the η phase which has formed at a two-layer stacking fault in the γ′ phase. Upper left is the HAADF–STEM image of the fault exhibiting characteristic ordering of intensity within the fault; Ni sublattice (green); Co (yellow) segregating to Ni sites; Ta (blue) segregating to the Wyckoff 2a sites; Al and Ti (red and magenta, respectively) segregating to the Wyckoff 2d sites. All elemental values are in at%. Scale bar, 0.5 nm.

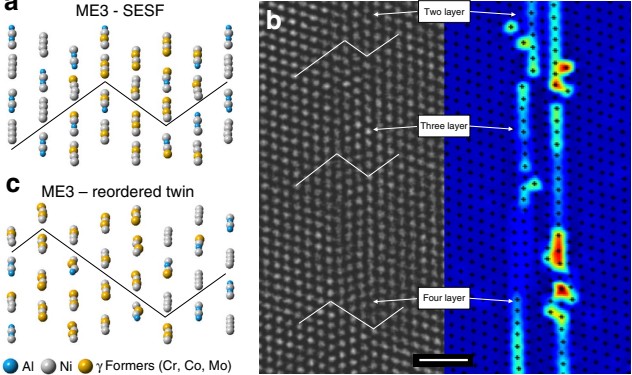

**Figure 4 | SESF to nanotwin transformation.** (a) DFT cell showing segregation along a SESF in ME3. (b) on the left is an experimental HAADF–STEM image of a two-layer SESF being sheared by two Shockley partials to form a three layer twin near a γ-γ′ interface. On the right is a centre of symmetry analysis of the HAADF–STEM image. (c) DFT cell showing the resulting twin from the process observed in (b). Scale bar, 1 nm.

γ-γ′ interface where the SESF has formed. These partials are able to enter the γ′ precipitate and shear along the SESF, with little energy penalty. The formation of a γ-like phase along the SESF removes nearest-neighbor violations, promoting further shearing by partials due to the subsequent lower energy barrier. Consequently, twins which shear through both γ and γ′ phases (Fig. 6c,d) can form, thereby defeating the effectiveness of the strengthening γ′ precipitates. This explains the high frequency of twins observed in ME3.

In the case of ME501, η phase formers (Nb, Ta and Ti) segregate to the SESF as shown in Fig. 6e. When Shockley partials interact with the fault at the γ-γ′ interface, the lack of nanotwinning implies that shearing along the fault is a rare occurrence and DFT calculations confirm that a very large energy barrier prevents these partials from shearing into the precipitate (Fig. 4a,b). Therefore, the η phase formation along isolated SESFs represents a new strengthening mechanism by limiting the formation of nanotwinning, greatly improving creep strength and high-temperature capabilities.

Conceptually, the distinct interface structures observed in both alloys are similar to the formation of interface 'complexions' which have been identified in the case of grain boundaries in several ceramic and metallic systems[27,28]. Complexions are 'interface-stabilized states'[29] that have a structure and composition different from that of the matrix and remain confined in the region where they form. This concept has recently been extended to 'linear complexions' along edge dislocations in a ferritic steel in which the core region of the dislocation reveals an FCC structure[29]. Common to this previous work is that these special structural states arise during thermal exposure, and thus appear to be thermodynamically stable when localized to the defects, and furthermore do not tend to grow (that is, thicken in the case of grain boundaries) with time. Thus, the stacking faults which are compositionally 'γ-like' in the case of alloy ME3, and form a one unit cell thick η phase in alloy ME501, both appear to have the attributes of 'complexions.' However, a new aspect to the present observations is that these special structural states have developed dynamically under applied stress during high-temperature deformation, as opposed to under static annealing, and thus may be accurately described as 'dynamic complexions' which have a direct and profound impact on the strength of the superalloy.

Therefore, the high-temperature strengthening mechanism discussed here has two important, beneficial aspects. First, the nucleation of a secondary ordered phase at stacking faults doubles the energy required to operate additional partial dislocations that are required for nanotwinning; and second, the inhibited segregation of γ formers to the stacking fault, thereby creating a local γ-like phase at the fault, which would promote nano-twinning. This 'phase transformation strengthening' mechanism operates in conjunction with the strengthening from secondary phase precipitates in γ-γ′ alloys, and is heretofore an unidentified strengthening mechanism. Indeed, this mechanism may be further manipulated through alloying and processing to further improve the high-temperature properties of next-generation superalloys for critical structural applications.

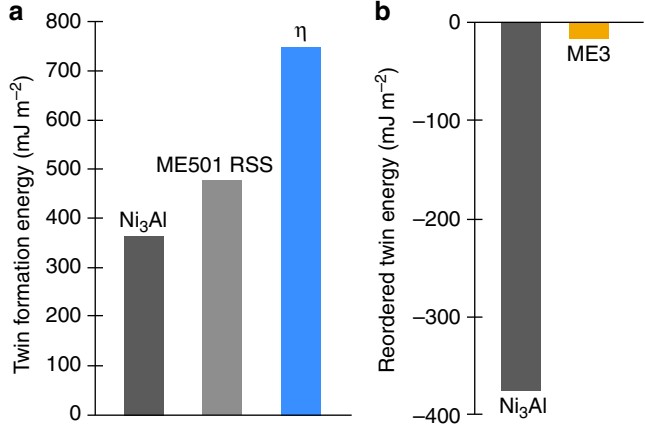

**Figure 5 | DFT calculations of SESF and nanotwin energies.** (**a**) Energetic cost of twin formation by shearing along a SESF before reordering in Ni₃Al, ME501 with a random solid solution, that is, no segregation (RSS), and ME501 where η has nucleated along the fault as observed experimentally. Note the relatively large energy cost to form a twin along a SESF with η phase. (**b**) Energy difference due to reordering after twin formation. The small difference found for ME3 suggests that the segregation of γ formers (Co, Cr, and Mo) replacing Ni and Al has removed nearest-neighbor violations in the precipitate near the fault, making twinning easier for ME3.

## Methods

**Microstructural characterization.** Single crystal analogues of the disk alloy ME3 (which has the following composition in wt%: 50.1% Ni, 20.6% Co, 13.0% Cr, 3.8% Mo, 2.1% W, 0.9% Nb, 2.4% Ta, 3.5% Al, 3.7% Ti, 0.04% C, 0.03% B and 0.05% Zr) and ME501, which has minor compositional variations relative to ME3 (increases in Ta, Hf and Nb and decreases in Co and Cr) were obtained from GE Global Research Center in the form of single, large castings after performing a heat treatment that formed a bimodal γ-γ′ microstructure. Before testing,

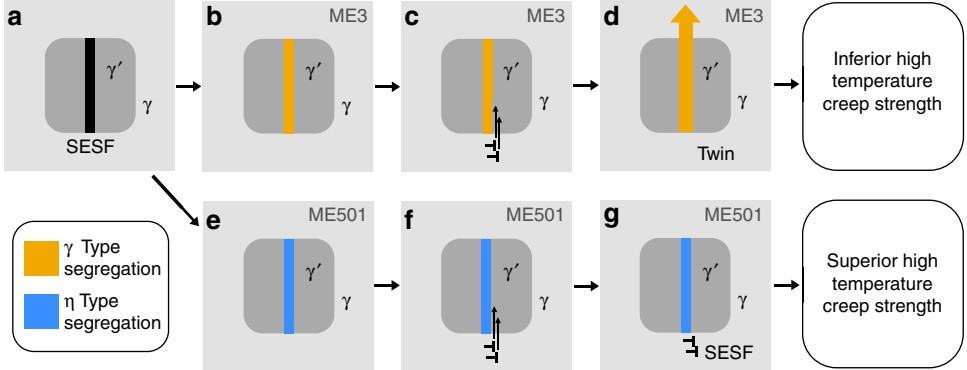

**Figure 6 | Phase transformation strengthening in ME501.** (**a**) Schematic of an isolated SESF in a γ′ precipitate. (**b**) γ formers (Co, Cr, and Mo) segregated along the SESF in ME3. (**c**) Two more dislocations have interacted at the γ-γ′ interface near the SESF in ME3. (**d**) Two more Shockley partials shear along the SESF forming a four-layer twin that is able to shear both the γ and γ′ precipitates. (**e**) η formers (Co, Ta, Ti, and Nb) segregated along the SESF in ME501. (**f**) Two more dislocations have interacted at the γ-γ′ interface near the SESF in ME501. (**g**) Given results in Fig. 5 the dislocations are not able to form a twin in ME501.

microstructure analysis on ME3 and ME501 was conducted to obtain volume fraction and average size for secondary precipitates. After polishing with SiC pads and 0.05 μm colloidal silica, each alloy was etched with a solution of 2 ml hydrofluoric acid, 30 ml nitric acid, and 50 ml lactic acid that preferentially etched the γ′ precipitates. Using an FEI Sirion scanning electron microscope (SEM), backscattered electron micrographs of the alloys' microstructures were obtained and then analyzed using ImageJ[30].

**Compression creep sample preparation.** Compression cuboids with a 1:1:2.5-dimension ratio were extracted from both bulk crystals using Electrical Discharge Machining (EDM). Monotonic compression creep tests were performed on both [001] and [110] oriented samples at 760 °C until 0.5% plastic strain was reached. For the ME3 tests a stress of 414 MPa was used while a stress of 552 MPa was used for the ME501 samples to obtain deformation under comparable strain rates ($5 \times 10^{-9}\,s^{-1}$). Linear variable displacement transducers were used on an MTS 810 compression cage to record the displacement of the compression cubes. Temperature was recorded using two K thermocouples. After the desired plastic strain was reached, the test was stopped and the specimen was very quickly returned to room temperature using a fan. All compression samples remained under load to preserve the deformation substructure that was present at the end of the test, and to minimize and changes to local structure and chemistry associated with dislocations and stacking faults.

**Electron microscopy characterization.** ECCI was used to obtain statistically significant occurrence values for nanotwinning in both alloys. Contrast from ECCI can provide information on local crystal distortions, for example, near-surface dislocations and faults such as twins[31]. Faults that appeared to shear both γ and γ′ phases were considered twins. After the oxidized layer was polished off of post-creep samples, TEM foils normal to the compression axis were extracted using an FEI Helios Nanolab DualBeam 600 focused ion beam. Samples were thinned at 5 kV and then further cleaned using a Fischione Nanomill. Thin foils were analyzed using BF and low-angle annular dark field detectors on an FEI Tecnai F20 STEM at 200 kV. Samples were also extracted normal to <110> orientations to view stacking faults edge-on using HAADF zone axis imaging conducted on a probe-corrected Titan³ 80–300 kV STEM. All atomic resolution HAADF–STEM images were corrected for scan distortions[32]. High spatial resolution EDX line scans were conducted at 300 kV using an image-corrected Titan³ 60–300 kV with a Super-X detector utilizing the Bruker Esprit software. The Super-X EDX detection system uses four silicon drift detectors that are located radially around the objective pole piece and specimen stage for improved collection performance.

Atomic resolution EDX maps were collected using a double aberration-corrected FEI Themis with Super-X EDX detector at 300 kV. The probe current was set to 50 pA and the dwell time to 25 μs per pixel with a total spectrum collection time of 641 s. Raw EDX data were extracted from the original spectral map and summed over a defined repeat unit based on the η phase crystal structure. The summed spectrum image was quantified using the Bruker Esprit software and experimentally determined Cliff–Lorimer k-factors from a solutionized ME501 sample.

**Density functional theory calculation details.** Energetics of twin formation in pure Ni₃Al, ME501 and ME3 alloys were studied using spin-polarized first principles calculations utilizing the Vienna Ab-initio Simulation Package[23]. Building on previous work on the energetics of SESF formation, supercells containing twin configuration both before and after the nearest-neighbor reordering process were relaxed using a quasi-Newtonian algorithm with electron exchange treated with the generalized gradient approximation including spin polarization in the formulation of Perdew et al.[33]. All calculations were performed using a $3 \times 7 \times 6$ Monkhorst-Pack k-point mesh with plane wave energy cutoffs at least 30% greater than the highest specified in the pseudopotentials used[34]. Brillouin zone integration was performed using first-order Methfessel–Paxton smearing with a smearing width of 0.2 eV (ref. 35). Computational cell size and shape were not allowed to relax for any but the pure Ni₃Al structures, but structures were relaxed internally until energy differences were $<10^{-4}$ eV. Additional calculations were performed to test the convergence of calculation cell size with regards to twin formation energy; the results of these calculations are displayed in Supplementary Fig. 7 and discussed in Supplementary Note 4.

**Data availability.** The data that support the findings of this study are available from the corresponding author upon request.

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

## Acknowledgements

T.M.S. acknowledges the support of GE Aviation for their support of this work through the GE University Strategic Alliance (USA) programme, and M.J.M. the support of the National Science Foundation and the DMREF programme under grant #1534826. N.A., W.W., B.D.E. and D.W.M. acknowledge the Center for Emergent Materials: an NSF MRSEC under award number DMR-1420451. W.W. acknowledges partial support from the Air Force Office of Scientific Research under Grant number FA9550- 12-1-0059. H.L.F. was supported by the Center for the Accelerated Maturation of Materials at OSU. FEI Company is acknowledged gratefully for providing access to, and assistance in operation of, the FEI Themis equipped with Super-X EDX technology in the Eindhoven NanoPort. This work was also supported in part by an allocation of computing time from the Ohio Super-computer Center. T.M.S. acknowledges Lee Casalena for helping create and format figures.

## Author contributions

T.M.S., B.D.E., N.A., W.W., D.W.M. and M.J.M. conceived and designed the experiments. T.M.S., B.D.E., N.A., A.C., and R.E.A.W. performed the experiments. T.M.S., B.D.E., N.A., R.E.A.W., H.L.F., W.W., D.W.M. and M.J.M. analyzed the data. A.W. and T.H. contributed reagents, materials and analysis tools. T.M.S., B.D.E., N.A. and M.J.M. wrote the paper.

## Additional information

**Competing financial interests:** The authors declare no competing financial interests.

**Publisher's note**: 

