## [Peer Review File · Nature Communications]

PEER REVIEW

Reviewers' comments:

Reviewer #1 (Remarks to the Author):

The paper reports on the deformation processes occurring in a Ni-superalloy designed for use in gas turbine engines. They use very novel and rapidly developing high resolution microscopy techniques to analyse the chemistry of individual atoms located at the core of these defects highlights important differences between two alloys showing significantly different creep performance. This build on previous work by the authors noting the segregation to stacking faults in superalloys but is a significant advance in a number of respects. Firstly the microscope techniques allow site-specific chemical analysis identifying different patterns of segregation resulting from the relatively minor differences in the composition of the alloys compared. Secondly observations of the pattern of deformation are observed and plausibly attributed to this chemical segregation, thirdly the difference is rationalised by the use of DFT modelling showing that the formation of the η phase on the fault resists the further widening of the original fault inhibiting what would other wise be a major deformation mechanism in the alloy.

Other work in the field is extensively referenced although, in fairness, this area has been principally pioneered by this group of researchers and, whilst representing a significant point in the development of these ideas, builds on their previous publications.

The quality of the data is first class and the analysis rigorous and well explained within the space constraints of the paper. The supplementary data is a useful addition in this respect. The pictures presented in Figure 3 represent unprecedented detail of the chemistry of stacking fault defects in metal-based systems.

Generally the paper is very lucid and well written. The paper could be improved in the following respects. The compositions of both alloys should be given explicitly. Also the effects of the additional γ' formers on the improved creep properties should be discussed (perhaps in the additional material) as not all the additional benefit can be attributed to the change in the nature of the faults. Figure 1 (b) and (c) are somewhat too small to show what is described in the text i.e. that the faults in ME501 are confined to the γ' precipitates.

I lingered rather over Figure 5, finding it somewhat confusing. The caption RSS (presumably random stacking fault) is not explained and the text in the figure caption awkward and not replicating the order on the graph. Also it would be more informative to compare the energetic penalty of widening the fault to a twin for the two alloys in addition to giving the effect of the

ordering on the alloy ME501. Ditto for the twin energy. The text in the paper describing this is much clearer but does not refer directly to Figure 5.

I would strongly recommend that the paper be published. The importance of the development of improved alloys for this discs in aeroengines is absolutely crucial and cannot be overemphasized. Their temperature capability dictates the design of the whole engine and hence it's efficiency. Improving our understanding of how these alloys work enables more intelligent and hence rapid alloy development. This paper broadens significantly the range mechanisms operative in alloys and helps explain how small changes in alloy composition can have a profound effect on the performance of these vital materials.

Catherine Rae

Reviewer #2 (Remarks to the Author):

In the paper entitled "Phase transformation Strengthening of High Temperature Superalloys" the authors report a novel strengthening mechanism in Ni-based super alloys by coupling atomic resolution microscopy (both structural and compositional characterization using HAADF-STEM and STEM-EDS) and density functional theory calculations. The results are indeed exciting and provide a revolutionary new insight into a novel strengthening mechanism for superalloys, at elevated temperatures, using state-of-the-art microscopy techniques. This strengthening mechanism has wide applicability to multiple superalloys and could potentially be used to design the next generation of superalloys.

This article deserves to be published. However, there are few points in the article that need further clarification - to the reviewer as well as to the general audience.

1. In the introduction section, the authors have assiduously listed their motivations for this work, and the related its importance to superalloys. However, in its current state the introduction appears to be specifically directed towards Ni-based superalloys. Since, Nature. Comm. is meant for a wider general audience, the text should be revised in the introduction section to give it a more generic "feel".
2. Judging from the line profiles from ME3 and ME502 (Figure 2), it appears that Ta, Nb and Ti in gamma prime vary within 2-4at%, 0.5-2at% and 5-6.5at% respectively, while Mo and Cr (gamma formers) vary as ~0.5-0.8% and ~1-1.5a% (very small change). So all changes are basically within ~2%. Therefore it is very intriguing that such a minor increase in Ta Nb and Ti one can dynamically nucleate the " η " phase. This indicates some sort of a threshold value in Ta, Nb and Ti concentrations, beyond which " η " phase forms during creep deformation. The authors

should comment more on this issue.

3. Although the authors have proposed the mechanism as a "phase transformation strengthening", their mechanistic description appears to be more of dynamic precipitation of " η " phase, which has been previously discussed in the literature, i.e. nucleation of a new phase aided by the segregation of solute atoms to dislocations. The authors may want to differentiate between the two concepts and clarify the differences between these two mechanisms, to better establish the novelty of their mechanism.

Other minor comments are :

- Include the zone axis along which the HRSTEMs in Figures 1(d) and 1(e) were recorded - seems like $\langle 110 \rangle$
- Include more information on the " η " phase: crystal structure, lattice parameter, space group etc.
- Include additional DFT parameters: cut-off energy, H-spacing, smearing width, Brillouin zone integration scheme, if the calculations were spin polarized or spin averaged, etc...

Timothy M Smith
Center for Electron Microscopy and Analysis
Department of Materials Science and Engineering
The Ohio State University
1305 Kinnear rd.
Columbus Oh, 43214
937-210-2611
smith.5881@osu.edu

Dear Editors and Reviewers,

We are very excited to have been given the opportunity to revise our manuscript, “Phase Transformation Strengthening of High Temperature Superalloys,” for Nature Communications. We have carefully considered the comments and concerns made by both reviewers and in the list below describe how each was addressed. We are also greatly appreciative of the effort and time it takes the reviewers to provide their helpful insights. Our responses to each of their comments are provided below.

Responses to Reviewer #1:

1.) **Comment:** The compositions of both alloys should be given explicitly.

Response: We agree that the compositions of both alloys should be given and have added a new table (shown below) to do so. The compositions are given in weight percent and is on page 1 of the supplemental.

Alloy	Ni	Co	Cr	Mo	W	Nb	Ta	Al	Ti	Hf	C	B	Zr
ME3	Bal.	20.6	13.0	3.8	2.1	0.9	2.4	3.5	3.4	0	0.05	0.03	0.03
ME501	Bal.	18.0	12.0	2.9	3.0	1.5	4.8	3.0	3.0	0.4	0.05	0.03	0.05

2.) **Comment:** The effects of the additional γ' formers on the improved creep properties should be discussed (perhaps in the additional material) as not all the additional benefit can be attributed to the change in the nature of the faults.

Response: Again, we agree with this statement as it's important to consider how the minor differences between the two alloys, not including twin formation, could have affected the creep properties. We added the text below in pages 4 and 5 of the supplemental to address this.

“Therefore, the increase in creep strength found in ME501 can be directly related to inhibition of creep by twinning. However, not all of the creep property differences between the two alloys can be explained solely through this phase transformation strengthening mechanism. The higher amount of γ' formers in ME501 led to a larger volume fraction of secondary precipitates (52% compared to 47% in ME3), and the finer microstructure may have also contributed to the creep strength differences. It is emphasized that the present comparison of twinning propensity is based on comparison of behavior at significantly different stress levels for the two alloys, in order to achieve similar creep rates. In this way, we have attempted to compensate for the volume fraction differences. Furthermore, while comparing coarse and fine microstructures in ME3, Unocic *et al.*⁹ found that the finer microstructure possessed better high temperature properties compared to the coarser microstructure. They attributed the improvement of properties to the smaller channel widths found in the fine microstructure, which inhibited dislocation motion. However, these tests were conducted on polycrystalline samples and the relationship between microstructure and creep properties is still not clear. Diologent and Caron¹⁰ found that increasing precipitate size, while keeping the volume fraction constant, improved the primary creep properties for single crystal AM1 and MC544. This improvement was attributed to a decrease of precipitate shearing dislocations in the larger precipitate samples. In this study, single crystal samples were tested, removing the complicating and confounding variables of grain boundaries, secondary phases along grain boundaries (ie carbides and borides), and grain size in order to more effectively explore the effects small alloying differences have on creep properties.”

3.) **Comment:** Figure 1(b) and (c) are somewhat too small to show what is described in the text i.e. that the faults in ME501 are confined to the γ' precipitates.

Response: We feel that it is important for the reader to discern that the faults are mainly confined to the precipitates in the case for ME501. Therefore, we have replaced Figure 1(c) with a higher magnification micrograph to better show this. We kept Figure 1(b) (ME3) the same because the goal of this image is to show the longer twins shearing both phases.

- 4.) **Comment:** I lingered rather long over Figure 5, finding it somewhat confusing. The caption RSS (presumably random stacking fault) is not explained and the text in the figure caption awkward and not replicating the order on the graph. Also it would be more informative to compare the energetic penalty of widening the fault to a twin for the two alloys in addition to giving the effect of the ordering on the alloy ME501. Ditto for the twin energy. The text in the paper describing this is much clearer but does not refer directly to Figure 5.

Response: We are thankful for this insightful comment, particularly since Figure 5 is key for communicating the essential conclusions of the. The caption in Figure 5 has been rewritten to be more clear (ie RSS is now defined and the information has been reordered to more closely correspond to the figure sequencing). We have also now referred to Figure 5 throughout the text to help the reader more fully understand what is being shown. The point of Figure 5(a) is to explicitly show the effect that the ordering found in ME501 has on the SESF-to-twin transformation. Hopefully, now that we have explained that RSS stands for a Random Solid Solution (i.e. an unsegregated fault), it should be clear that the ordered eta phase along a SESF creates a larger energy barrier for the fault to extend into a twin compared to an unsegregated/unordered fault. The reason that ME3 is not included in Figure 5(a) as well as why ME501 is not included in 5(b) is because we felt that doing so did not add anything new to our conclusions and made the figures harder to understand. Below is the new caption to Figure 5.

“(a) Energetic cost of twin formation by shearing along a SESF prior to reordering in Ni_3Al , ME501 with a random solid solution, i.e. no segregation (RSS), and ME501 where η has nucleated along the fault as observed experimentally. Note the relatively large energy cost to form a twin along a SESF with η phase. (b) Energy difference due to reordering after twin formation. The small difference found for ME3 suggests that the segregation of γ formers (Co, Cr, and Mo) replacing Ni and Al has removed nearest-neighbor violations in the precipitate near the fault, making twinning easier for ME3.”

Again, we are grateful for all of the insightful comments/concerns and your help with making this a better manuscript.

Responses to Reviewer #2:

1.) **Comment:** In the introduction section, the authors have assiduously listed their motivations for this work, and the related its importance to superalloys. However, in its current state the introduction appears to be specifically directed towards Ni-based superalloys. Since, Nature. Comm. is meant for a wider general audience, the text should be revised in the introduction section to give it a more generic "feel".

Response: On their website, Nature Communications states the following, "*Nature Communications* is an open access, multidisciplinary journal dedicated to publishing high-quality research in all areas of the biological, physical, chemical and Earth sciences. Papers published by the journal represent important advances of significance to specialists within each field." Since it is stated that the manuscripts published in Nature Communications represent important advances to specialists within each field we feel that our introduction is very broad with respect to superalloy development and fits very well with the subject of our study and what's expected for this journal. We believe that the results of this work are generally applicable to Ni base superalloys, but do not presently have basis to extend the conclusions beyond these alloys. Nevertheless, we do refer to analogous segregation/transformation behavior observed in Co-base alloys by Titus and co-workers.

2.) **Comment:** Judging from the line profiles from ME3 and ME501 (Figure 2), it appears that Ta, Nb and Ti in gamma prime vary within 2-4at%, 0.5-2at% and 5-6.5at% respectively, while Mo and Cr (gamma formers) vary as ~0.5-0.8% and ~1-1.5a% (very small change). So all changes are basically within ~2%. Therefore it is very intriguing that such a minor increase in Ta Nb and Ti one can dynamically nucleate the "η" phase. This indicates some sort of a threshold value in Ta, Nb and Ti concentrations, beyond

which " η " phase forms during creep deformation. The authors should comment more on this issue.

Response: This is an excellent observation and one that we should address in our manuscript. The following text was added to page 6 of the manuscript. "The small amount of segregation needed to nucleate the η phase along the SESF implies that a threshold may exist where η phase is promoted over γ' . In fact, one study found that when the C content was lowered from 0.15 wt% to 0.08 wt% in IN792, thereby increasing the content of carbide formers (Ta, Nb, and Ti) in the bulk alloy, that η phase would begin to form¹³. It may be that the composition of ME501 is approaching this threshold."

3.) Comment: Although the authors have proposed the mechanism as a "phase transformation strengthening", their mechanistic description appears to be more of dynamic precipitation of " η " phase, which has been previously discussed in the literature, i.e. nucleation of a new phase aided by the segregation of solute atoms to dislocations. The authors may want to differentiate between the two concepts and clarify the differences between these two mechanisms, to better establish the novelty of their mechanism.

Response: We have attempted to make this distinction, and further clarified this distinction in the revised text (pg. 13 of the manuscript). Indeed, the term "phase transformation strengthening" can be used to describe cases where transformations occur in the presence of externally imposed stresses, not necessarily in the presence of specific defects such as dislocations and stacking faults and not necessarily at elevated temperatures. We have therefore also made reference to "dynamical complexions" as an alternative view of the observed mechanism since the localized nature of these phase transformations at the stacking faults is compatible with the relatively new concept of "complexions." Full referencing to these concepts have been provided in the text and is shown below.

“Conceptually, the distinct interface structures observed in both alloys are similar to the formation of interface “complexions” which have been identified in the case of grain boundaries in several ceramic and metallic systems.^{27,28} Complexions are “interface-stabilized states”²⁹ that have a structure and composition different from that of the matrix and remain confined in the region where they form. This concept has recently been extended to “linear complexions” along edge dislocations in a ferritic steel in which the core region of the dislocation reveals an FCC structure²⁹. Common to this previous work is that these special structural states arise during thermal exposure, and thus appear to be thermodynamically stable when localized to the defects, and furthermore do not tend to grow (i.e. thicken in the case of grain boundaries) with time. Thus, the stacking faults which are compositionally “ γ -like” in the case of alloy ME3, and form a one unit cell thick η phase in alloy ME501, both appear to have the attributes of “complexions.” However, a new aspect to the present observations is that these special structural states have developed dynamically under applied stress during high temperature deformation, as opposed to under static annealing, and thus may be accurately described as “dynamic complexions” which have a direct and profound impact on the strength of the superalloy.”

- 4.) Comment: Include the zone axis along which the HRSTEMs in Figures 1(d) and 1(e) were recorded - seems like $\langle 110 \rangle$

Response: We agree with this sentiment and have included the zone axis for all of the HRSTEM images in Figure 1(d and e) and Figure 2(a and b).

- 5.) Comment: Include more information on the " η " phase: crystal structure, lattice parameter, space group etc.

Response: We have addressed this comment by adding the following information in our manuscript on page 6. “Bulk η phase possesses a hexagonal $D0_{24}$ crystal structure ($P6_3/mmc$, $a=.5096$, $c=.8304$, $\alpha=\beta=90^\circ$, $\gamma=120^\circ$), the same found locally along a SESF^{8,9}”

6.) Comment: Include additional DFT parameters: cut-off energy, H-spacing, smearing width, Brillouin zone integration scheme, if the calculations were spin polarized or spin averaged, etc...

Response: It was a mistake on our part that this wasn't included in the first draft. The following paragraph has been added to the end of the methods section of the manuscript (page 16 and 17).

“Energetics of twin formation in pure Ni₃Al, ME501, and ME3 alloys were studied using spin-polarized first principles calculations utilizing the Vienna Ab-initio Simulation Package (VASP).²³ Building on previous work on the energetics of SESF formation, supercells containing twin configuration both before and after the nearest-neighbor reordering process were relaxed using a quasi-Newtonian algorithm with electron exchange treated with the generalized gradient approximation (GGA) including spin polarization in the formulation of Perdew, Burke, and Ernzerhof²³. All calculations were performed using a 3×7×6 Monkhorst-Pack k-mesh with plane wave energy cutoffs at least 30% greater than the highest specified in the pseudopotentials used³⁴. Brillouin zone integration was performed using first-order Methfessel-Paxton smearing with a smearing width of 0.2 eV³⁵. Computational cell size and shape were not allowed to relax for any but the pure Ni₃Al structures, but structures were relaxed internally until energy differences were less than 10⁻⁴ eV. Additional calculations were performed to test the convergence of calculation cell size with regards to twin formation energy; the results of these calculations can be found in the supplementary materials. “

Again, we are grateful for all of the insightful comments/concerns and your help with making this a better manuscript.

Reviewers' Comments:

Reviewer #1 (Remarks to the Author):

I have looked at the revised manuscript and think the authors have responded fully to the comments made. I am happy to repeat my recommendation that the paper be published.

Reviewer #2 (Remarks to the Author):

The authors have satisfactorily addressed the comments/suggestions associated with the original submission. The reviewer is satisfied with the modifications and the revised manuscript. Therefore, the reviewer would like to recommend that this article be accepted for publication.

Timothy M Smith
Center for Electron Microscopy and Analysis
Department of Materials Science and Engineering
The Ohio State University
1305 Kinnear rd.
Columbus Oh, 43214
937-210-2611
smith.5881@osu.edu

Dear Editors,

We are very excited to have been given the opportunity to revise our manuscript, “Phase Transformation Strengthening of High Temperature Superalloys,” for Nature Communications. We are also greatly appreciative of the effort and time it takes the reviewers to provide their helpful insights. As can be seen by the reviewer’s comments below, all of their previous concerns have been adequately addressed in the previously revised manuscript.

Reviewer #1:

I have looked at the revised manuscript and think the authors have responded fully to the comments made. I am happy to repeat my recommendation that the paper be published.

Reviewer #2:

The authors have satisfactorily addressed the comments/suggestions associated with the original submission. The reviewer is satisfied with the modifications and the revised manuscript. Therefore, the reviewer would like to recommend that this article be accepted for publication.

Again, we are grateful for all of the insightful comments/concerns and your help with making this a better manuscript.

Sincerely,

Timothy Smith